# Mass spectrometry imaging for biosolids characterization to assess ecological or health risks before reuse

Claire Villette [1], Loïc Maurer [2], Julie Zumsteg [1], Jérôme Mutterer [3], Adrien Wanko [2] & Dimitri Heintz [1] ✉

Biosolids are byproducts of wastewater treatment. With the increasing global population, the amounts of wastewater to be treated are expanding, along with the amounts of biosolids generated. The reuse of biosolids is now accepted for diversified applications in fields such as agriculture, engineering, agro-forestry. However, biosolids are known to be potential carriers of compounds that can be toxic to living beings or alter the environment. Therefore, biosolid reuse is subject to regulations, mandatory analyses are performed on heavy metals, persistent organic pollutants or pathogens. Conventional methods for the analysis of heavy metals and persistent organic pollutants are demanding, lengthy, and sometimes unsafe. Here, we propose mass spectrometry imaging as a faster and safer method using small amounts of material to monitor heavy metals and persistent organic pollutants in different types of biosolids, allowing for ecological and health risk assessment before reuse. Our methodology can be extended to other soil-like matrices.

Biosolids are the residual semisolid material produced by wastewater treatment plants (WWTP) after the treated water is removed[1]. The terms sewage sludge and biosolid are sometimes used interchangeably, but sludge designates a nontreated liquid[2,3], while biosolid refers to a sludge processed through filtering, aerobic/anaerobic digestion, alkaline stabilization, thermal drying, acid oxidation/disinfection, composting, etc., resulting in stabilized organic solids that can be safely reused with benefits in terms of nutrients, energy, etc.[4]. With the increasing global population, measured at 7.3 billion people in 2015 and expected to reach 8.5 billion by 2030[5], the amount of wastewater to be treated increases, as the amount of resulting biosolids. In 2019, the amounts of biosolids (dry weight) produced were 371,000 tons in Australia[6], 7.8 million tons in China[7], 3 million tons in European countries[8], and 4.75 million tons in the USA[9]. The cost of sewage sludge management represents approximately 50% of the running costs of WWTP[10]; therefore, the economic aspects of biosolids have raised. The increase in biosolid reuse is a solution to overcome these costs, and multiple recycling possibilities are emerging[10]. For review, an inventory of worldwide biosolid production and use was produced in 2008[11], and a review of the different types of reuse was proposed recently[10]. Biosolids are nutrient-rich products that are appreciated for agricultural land application as a fertilizer, reclamation sites (to establish sustainable vegetation, reduce bioavailability of toxicants in soils, control erosion, regenerate soil layers), forestry applications (to promote timber growth, allowing quicker harvest of an important natural resource), engineering applications (construction of roads, production of bricks, ceramics, lightweight aggregates, cements), and use in lawns and home gardens (biosolids meeting the most stringent pollutant, pathogen and vector attraction reduction requirements).

Biosolids have long been described as potential carriers of hazardous compounds that might be toxic to humans and ecosystems. The first compounds to be monitored in environmental matrices were heavy metals (HMs), and the list of priority compounds was implemented over time with contaminants of emerging concern[12,13] and

[1]Plant Imaging & Mass Spectrometry (PIMS), Institut de biologie moléculaire des plantes, CNRS, Université de Strasbourg, 12 rue du Général Zimmer, 67084 Strasbourg, France. [2]Université de Strasbourg, CNRS, ENGEES, ICube UMR 7357, F-67000 Strasbourg, France. [3]Microscopie et Imagerie Cellulaire, Institut de biologie moléculaire des plantes, CNRS, Université de Strasbourg, 12 rue du Général Zimmer, 67084 Strasbourg, France. ✉e-mail: dimitri.heintz@ibmp-cnrs.unistra.fr

persistent organic pollutants (POPs). For example, per- and poly-fluoroalkyl substances (PFASs) have recently attracted attention as they are persistent compounds[14,15] and would disseminate in the environment if present in biosolids to be applied on agricultural land. To assess the ecotoxicological risk associated with land application of biosolids, the usual risk quotients strategy is used[16–18] in association with local legislations that impose thresholds on some priority compounds[19]. This topic is of great interest to the scientific community, as several reviews dealing with biosolids recently detailed their effects on soil properties[20]; the life cycle and risk assessment of biosolids; their POPs, metallic and pathogenic contents; and the processes to make biosolids safe for reuse[1,21,22].

To search for contaminants in biosolids, inductively coupled plasma (ICP) and gas and liquid chromatography (GC and LC) coupled to mass spectrometry (MS) are routinely used[23], but necessitate laborious sample preparation to extract the compounds of interest from semisolid materials. We propose the use of matrix-assisted laser desorption ionization mass spectrometry imaging (MALDI-MSI), which reduces the time and effort required for sample preparation while achieving the identification of HMs and a hundred of POPs from a single sample, including emerging compounds known to be difficult to analyze with usual methods. As already demonstrated, MALDI-MSI is a powerful tool in the context of environmental sciences[24–27] but has not yet been applied to the evaluation of biosolid content. MALDI-MSI is used to image the distribution of various biological molecules in tissue sections without antibodies, staining, or complex pretreatment steps. It provides both spatial and molecular information for hundreds of ionized analytes based on their molecular masses in a single measurement, enabling the acquisition of ion density maps[28,29]. The simplicity of sample preparation, rapid spectral acquisition, high sensitivity, and relative tolerance to impurities make MALDI-MSI attractive in various scientific fields including materials science, forensics and paintings analysis[30–36]. This technique is helpful for the high-throughput direct detection of diverse analytes. Thus, MALDI-MSI is a suitable tool for investigating the distribution of analytes in biosolids. The images obtained are not expected to show the native distribution of the analytes of interest in biosolids, but rather to give additional information which can help go further in the annotation process. Indeed, a solvent-forced migration is applied during sample deposition, providing a spatial distribution, which is also linked to the physico-chemical properties of the compounds of interest.

The social acceptance of biosolid reuse requires a methodology to control their quality. An effortless methodology to analyze biosolid samples rapidly does not yet exist. To determine the presence of HMs and POPs in biosolids, several methods are necessary, which are not currently adapted for high throughput and cost-efficient analysis to ensure a wide screening of the key components in environmental contaminants.

In this work, we propose a simplified and safer method using MALDI-MSI to evaluate the contents of biosolids before reuse. We analyze POPs and HMs contents in nine different biosolid samples resulting from varied wastewater treatments. Using MALDI-MSI, we detect pollutants families which are not easily detected with conventional methods (LC/GC-MS). Moreover, we demonstrate that the distribution of the pollutants in the samples after deposition is linked to physico-chemical properties of the compounds, namely soil adsorption and water solubility. Here, we show that MALDI-MSI is a suitable tool for the nontargeted analysis of POPs and HMs in non-cohesive materials like biosolids.

## Results
### Sampling strategy
The sampling strategy consisted of choosing very diversified biosolids that are representative of the most used WWTP (Table 1, Supplementary Table 1 and Supplementary Fig. 1), although the presented

sampling is not exhaustive due to the variety of treatments existing. Samples were obtained from constructed wetlands and activated sludge, collecting wastewater from small villages to very large cities (160–1,000,000 people equivalent), with a wide range of flow rates (190–305,322 $m^3$/day) and different sludge treatment processes, providing biosolids of different ages (2 days–12 years), subjected or not to thermal treatment (37–180 °C). The biosolids sampled were intended for agricultural use, but the proposed method also applies to biosolids intended for other forms of reuse. This wide diversity was chosen to show that the proposed analytical method is suitable for a wide range of biosolids.

### MALDI-MSI simplifies the analysis of biosolids thanks to conductive tape
Currently, biosolid content is investigated using at least three different methods to obtain information necessary for its reuse following legislation. ICP is needed to determine the HMs content, and LC/GC-MS are used to evaluate the presence of POPs. To perform these analyses, hundred grams of biosolids are needed, and the analyses take time and several different instrumental setups (Fig. 1a). We propose to use MALDI-MSI to analyze HMs and POPs at once from only 1 g of homogenized biosolids using a unique method that takes less than 2 h of sample preparation and avoids dangerous steps (Fig. 1a). Only one instrument is necessary, and the sample preparation and acquisition time are dramatically reduced as compared to conventional methods. Moreover, the data obtained are not limited to the list of suspected contaminants and can be further analyzed, as they are acquired using a nontargeted approach.

The analysis of biosolids with MALDI-MSI is not straightforward when using usual sample preparation techniques. Indeed, these samples are powdery and tend to fall to pieces when applied on a classical MALDI slide (Fig. 1b and Supplementary Fig. 2). Samples used for MALDI-MSI must be flat and strongly adhere to the slide to stay attached in the vacuum of the mass spectrometer source where the sample is placed for the analysis. Several authors reported the utility of a conductive tape as a solution for fragile biological tissues that are destroyed by cryo-sectioning or handling[37,38]. We compared several types of conductive tapes and obtained the best results with a copper-based tape recently introduced by the Saito group[39] (Supplementary Table 2). The use of a conductive tape avoided sample crumbling (Fig. 1b and Supplementary Fig. 2). Moreover, it allowed for us to test several sample thicknesses (Fig. 2) and we were able to demonstrate that (1) the samples remained intact and (2) we obtained signals in all samples regardless of the thickness.

### Analysis of HMs and POPs in biosolids using MALDI-MSI
The samples were irradiated by a laser every 50 μm, which resulted in the ionization of the compounds contained in the samples at the location irradiated by the laser. The mass-to-charge (m/z) ratios of these ions were recorded in mass spectra. The obtained mass spectra showed the content of the samples using a nontargeted analysis. A mass spectrum was recorded for every 50 × 50 μm pixel, resulting in spatially resolved images, on which the localization and abundance of a compound of interest can be visualized. The mass spectra obtained were used to annotate the sample content with HMs and POPs, giving names to the measured m/z ratios.

HMs have the capability to combine with chloride ions released by the MALDI matrix; which allows their ionization and analysis using mass spectrometry. The very specific isotopic patterns obtained from the ionization of chlorinated metals allows for their confirmed identification (level 1 of the Schymanski classification[40]): zinc, copper (Fig. 3), iron, lead, magnesium and nickel (Supplementary Fig. 3) were successfully detected in the samples.

From the nine samples used in this study, 499 POPs were annotated (level 3), among which 83 annotations could be confirmed using

**Table 1 | Diversity of the biosolid samples used for the study**

|  | Age | Capacity (people equivalent) | Inlet flowrate (m³/day) | Type |  | Sludge treatment |
|---|---|---|---|---|---|---|
| 1-Falkwiller | 12 years | 1450 | 450 | Constructed wetland | Aerobic | Drying on tarp |
| 2-Lutter | 12 years | 970 | 1080 | Constructed wetland | Aerobic | Drying on tarp |
| 3-Montiers | 9 years | 160 | 190 | Constructed wetland | Aerobic | Drying on tarp |
| 4-Sarrebourg | 6 months | 37,000 | 8616 | Activated sludge | Aerobic | Filter press drying + lime stabilization |
| 5-Hartzwiller | 6 months | 5040 | 1010 | Activated sludge | Aerobic | Filter press drying without stabilization |
| 6-Laneuville | 6 months | 9133 | 1720 | Activated sludge | Aerobic | Filter press drying + lime stabilization |
| 7-Wantzenau | 2 days | 1,000,000 | 305,322 | Activated sludge | Anaerobic | Thickening, digestion (biogas, 37 °C) + rotary drying |
| 8-Meistratzheim | 30 days | 204,550 | 22,194 | Activated sludge | Anaerobic | Digestion (biogas, 37 °C) + thermal drying (180 °C) |
| 9-Sausheim | 2 days | 490,000 | 126,115 | Activated sludge | Aerobic | Physico-chemical treatment, digestion (biogas, 37 °C) + rotary drying |

LC coupled to high resolution tandem mass spectrometry (HRMS/MS) or GC coupled to triple quadrupole tandem mass spectrometry (TQ-MS/MS, level 3 and 1). A hundred of POPs could be successfully annotated from a single sample, with varied spatial distributions and abundances (Fig. 4b and Supplementary Table 3). As expected, the biosolids which underwent a hard thermal treatment (180 °C) display a reduced number of POPs (Meistratzheim, 28 POPs annotated in MALDI-MSI). The POPs were classified according to the Norman Database classification: 13 families were annotated in biosolids (Fig. 4a), with four families that were not detected using conventional methods: glycol ethers, phthalates, surfactants, and PFASs (Fig. 4c).

### The spatial distribution of POPs in biosolids is linked to their physico-chemical properties

In this study, the native localization of POPs in biosolids was not of interest, and the sample preparation and deposition process revealed varied spatial distributions. The diversity in the POPs spatial distributions among a single sample could be explained through physico-chemical parameters, representative examples are shown in Fig. 5. The properties and environmental fate/transport parameters of POPs were recovered from the CompTox Chemicals Dashboard (Supplementary Table 4) and compared to the spatial distribution observed in the samples. Two parameters could be linked to the distribution of the POPs: water solubility and soil adsorption, with more hydrophilic compounds localized in the periphery of the samples, while more hydrophobic compounds that also show a higher soil adsorption coefficient could be found either in the middle of the sample or everywhere on the biosolid deposit. These distributions are due to an intentional solvent-forced migration of the compounds during sample deposition, which brings additional information on the detected compounds that can be used to validate tentative identifications in a nontargeted approach. The obtained samples were analyzed with an approach comparable to TLC-imaging, where migration adds information. The specific distributions observed were not reproduced when using 100% $H_2O$ for sample preparation and deposition, and analytes were not sufficiently extracted from biosolids to be correctly detected in the absence of a solvent (Supplementary Fig. 4).

### Quantification of HMs and POPs in biosolids using mass spectrometry imaging

Thanks to the accumulation of several microliters of sample on the same spot, HMs and POPs were detected and quantified in biosolids (Supplementary Figs. 5 and 6). An example is given for copper (HM) and telmisartan (POP). A standard curve was established by spotting the analytical standards onto a glass slide. For each biosolid sample,

one replicate was dedicated to quantification and spotted with 10 μg of copper or 15 pmol of telmisartan. The background signal from the sample itself was deduced from the 10 μg/15 pmol signal. The signal obtained from the sample spot was compared to the signal obtained from the 10 μg/15 pmol spot of the standard curve on the slide, and extinction factors were calculated for each sample on the same principle as the matrix effect in LC. Therefore, standard curves were derived for each sample, and the final quantity could be calculated, which was comparable to the quantities obtained from conventional methods. Copper was quantified at 91.48 mg/kg using MALDI-MSI and 92.2 mg/kg using ICP, while telmisartan was quantified as low as 4 μg/kg using MALDI-MSI and 5.7 μg/kg in LC-HRMS/MS.

## Discussion

MSI is extensively used for biological applications, but it also proved useful in other fields: in forensics to reconstitute fingerprints[34] or search for drugs of abuse in low amounts of hair[35]; in art to study old paintings without invasive protocols[33], in materials analysis to search for surface irregularities[36]. As in the cited applications, the present study demonstrates the ability of MSI to bring additional information on the samples of interest without a biological background. The authors have no interest in the original localization of the compounds in the native biosolid samples, but take advantage of the solvent-forced migration to (1) release the compounds of interest from possible interactions with the sample and make them available for ionization (2) concentrate low abundant analytes which could not be analyzed otherwise in specific areas of the samples; (3) help validating tentative identifications thanks to physico-chemical properties.

### A safe and effortless method for the characterization of biosolids before reuse

A growing number of biosolids is used to make sustainable materials and for engineering[10], increasing the demand for biosolid contaminant evaluation over time. In this context, the need for fast and simple biosolid characterization is growing. Usually, HMs are analyzed using ICP, and POPs are analyzed using LC/GC-MS, which necessitates the use of dangerous chemicals and high temperatures for the extraction of the compounds of interest and might take 2 to 3 working days[41]. We propose MALDI-MSI to simplify the detection of HMs and POPs using a single method, with faster and safer sample preparation from lower amounts of biosolids: only 1 g of sample material is needed, while several hundred grams might be necessary to perform multiple conventional methods. We deliberately chose very distinct biosolids from the most representative WWTP worldwide (even though other types of processes might occur), displaying different physico-chemical and

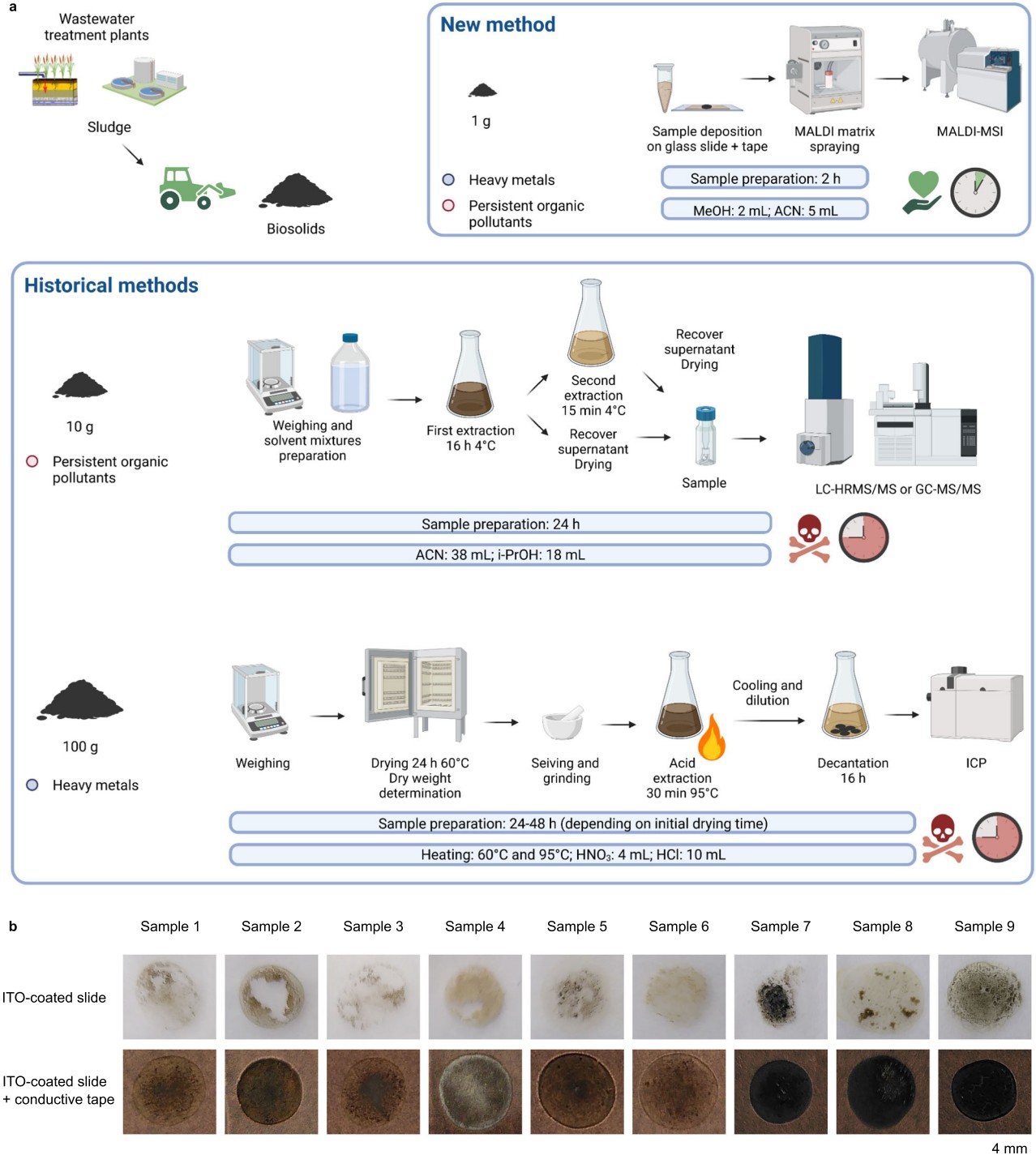

**Fig. 1 | Matrix assisted laser desorption ionization mass spectrometry imaging (MALDI-MSI) simplifies the analysis of biosolids thanks to conductive tape.** **a** Currently, two to three different methods are used to analyze heavy metals (HMs) and persistent organic pollutants (POPs) in biosolids: inductively coupled plasma (ICP) for HMs and liquid and/or gas chromatography (GC/LC) coupled to mass spectrometry (MS) for POPs. MALDI-MSI analysis allows for the detection of HMs and POPs with a single technique. **b** Biosolid samples deposited on indium tin oxide (ITO)-coated slides used for MALDI-MSI analysis tend to fall to pieces, while the use of a conductive tape allows for sufficient sample adhesion to the slide. Three independent experiments were performed with similar results. MeOH methanol, ACN acetonitrile, i-PrOH isopropanol, HNO₃ nitric acid, HCl hydrochloric acid.

metabolic profiles. In doing so, we demonstrate that this method is suitable for the analysis of varied types of biosolids and could be extended to other powdered materials (soil, sand, compost, etc.).

**Conductive tape is essential for proper sample fitting**
The good fitting of samples on the slide was allowed by a conductive tape, previously described to ease the preparation of fragile samples

such as plant tissues[39]. Several conductive tapes are available, and different authors used either copper-based or carbon tape for varied applications[37,38]. However, the use of a conductive tape for the manual deposition of powdered samples as biosolids was never described. The chosen copper tape has the lowest electrical resistance value compared to others (Supplementary Table 2). It is obvious that low electrical resistance will benefit to sample conductivity and end up with

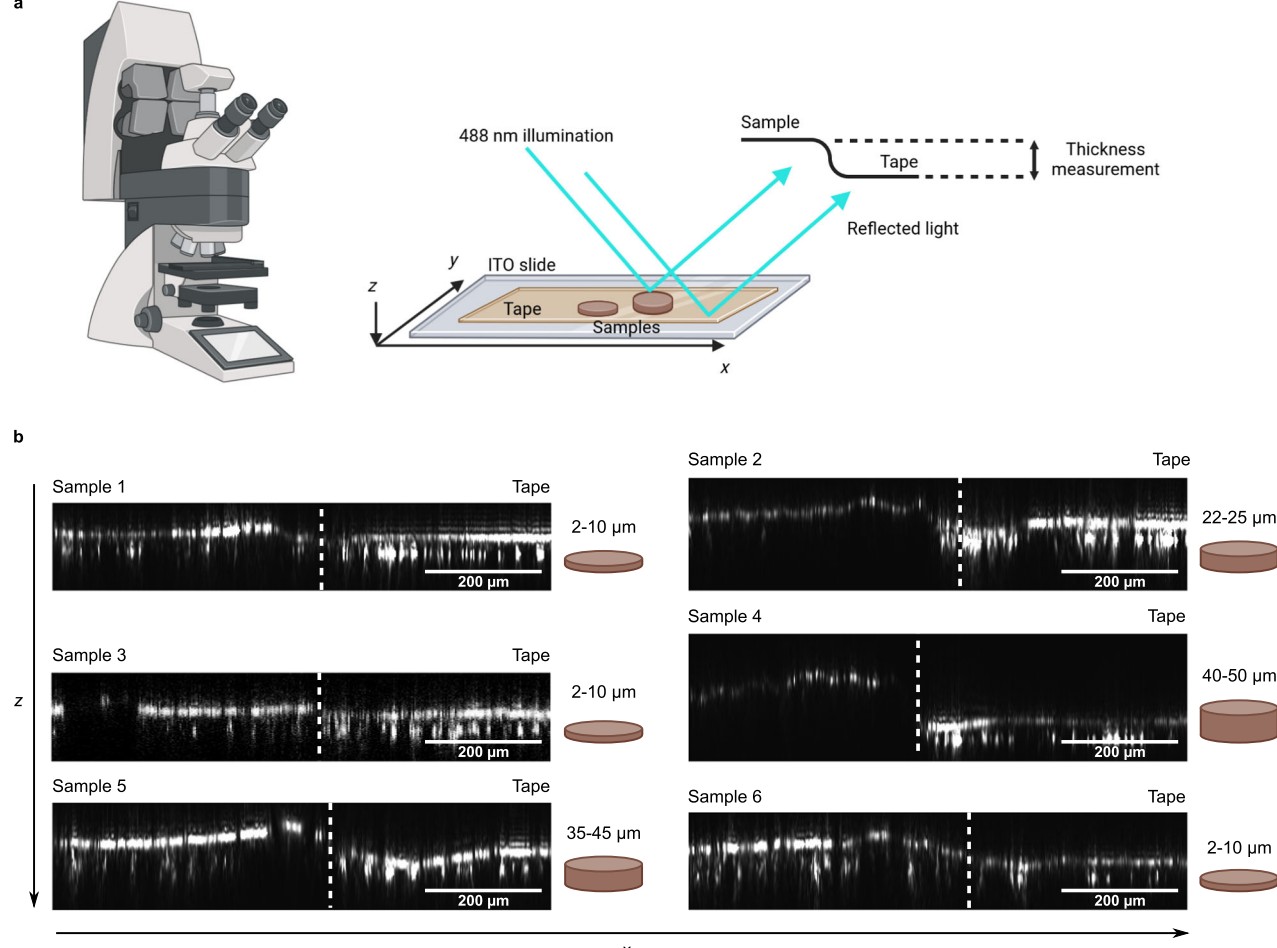

**Fig. 2 | Measurement of sample thickness using laser scanning microscopy.**
**a** Different sample thicknesses were tested, and the sample thickness was measured as the difference in height between the surface of the biosolid samples and the surface of the conductive tape. **b** Biosolid samples deposited on the conductive tape showed various thicknesses with a slight accumulation of material on the periphery of the samples. Three independent experiments were performed with similar results.

better signal intensity. The successive deposition of small quantities of samples followed by drying steps was successful in keeping the samples attached.

## Efficiency of MALDI-MSI to detect HM and POPs in biosolids

Biosolid analysis using MALDI-MSI allows for the annotation of hundreds of POPs, including compounds that are known to be difficult to analyze if not using specific methods. Indeed, glycol ethers, phthalates, surfactants and PFASs were annotated using MALDI-MSI but were not detected using conventional methods. Based upon environmental persistence, human toxicity, evidence of bioaccumulation in humans and the environment, evidence of ecotoxicity and the number and quality of studies focused on POPs internationally, PFASs and phthalates are considered chemicals of concern, which MALDI-MSI was able to reveal. The ability of the MALDI technique to detect perfluorinated chemicals was demonstrated by ref. [42], who detected perfluorooctanesulfonic acid in mouse tissues. This is an important outcome, as PFASs have real effects on human health (cancer, immune function, metabolic outcomes, neurodevelopment) and show very diverse pathways for human exposure[43].

Soil contamination with HMs is a major concern, as they are very difficult to eliminate: HMs are not biodegradable, and their persistence in soil is much longer than that of any other reactive component in terrestrial ecosystems. Some HMs are considered essential micronutrients for plant growth[44] (e.g., Cu and Zn), but elevated concentrations of these compounds are toxic to food crops, domestic animals and humans. The presence of HMs in biosolids can be due to the collection of urban and road runoff water or industrial effluents by WWTP. The presence of HMs is an important parameter to evaluate before biosolid reuse, as they were reported to be potential threats to human health[45] (arsenic is carcinogenic; cadmium is probably carcinogenic, teratogenic and embryotoxic; mercury is teratogenic). HMs can be analyzed by MALDI[46] and were identified in biological tissue slices by MALDI-MSI[47]. The use of a specific chlorinated matrix allows for the ionization of HMs species by the addition of one or several atoms of chloride. To our knowledge, this method was never used for the characterization of HMs in biosolid samples.

Another focus for the reuse of biosolids is their pathogen content and the associated risk to human health and the environment. Pathogen resistance is highly monitored and could occur due to the use of biosolids for land application[48]. As demonstrated by its wide use in hospitals for daily microbiology[49], MALDI proves very useful for the identification of pathogens. Indeed, metabolomics of human pathogens is increasingly documented with several databases[50–53] available for the annotation of human disorders[54,55]. A similar technology could be developed using MALDI-MSI to assess the pathogen content of biosolids. As a proof of concept that necessitates further development, we annotated biosolids using pathogen-specific databases and were able to detect two metabolites described as pathogen signatures (Supplementary Fig. 7).

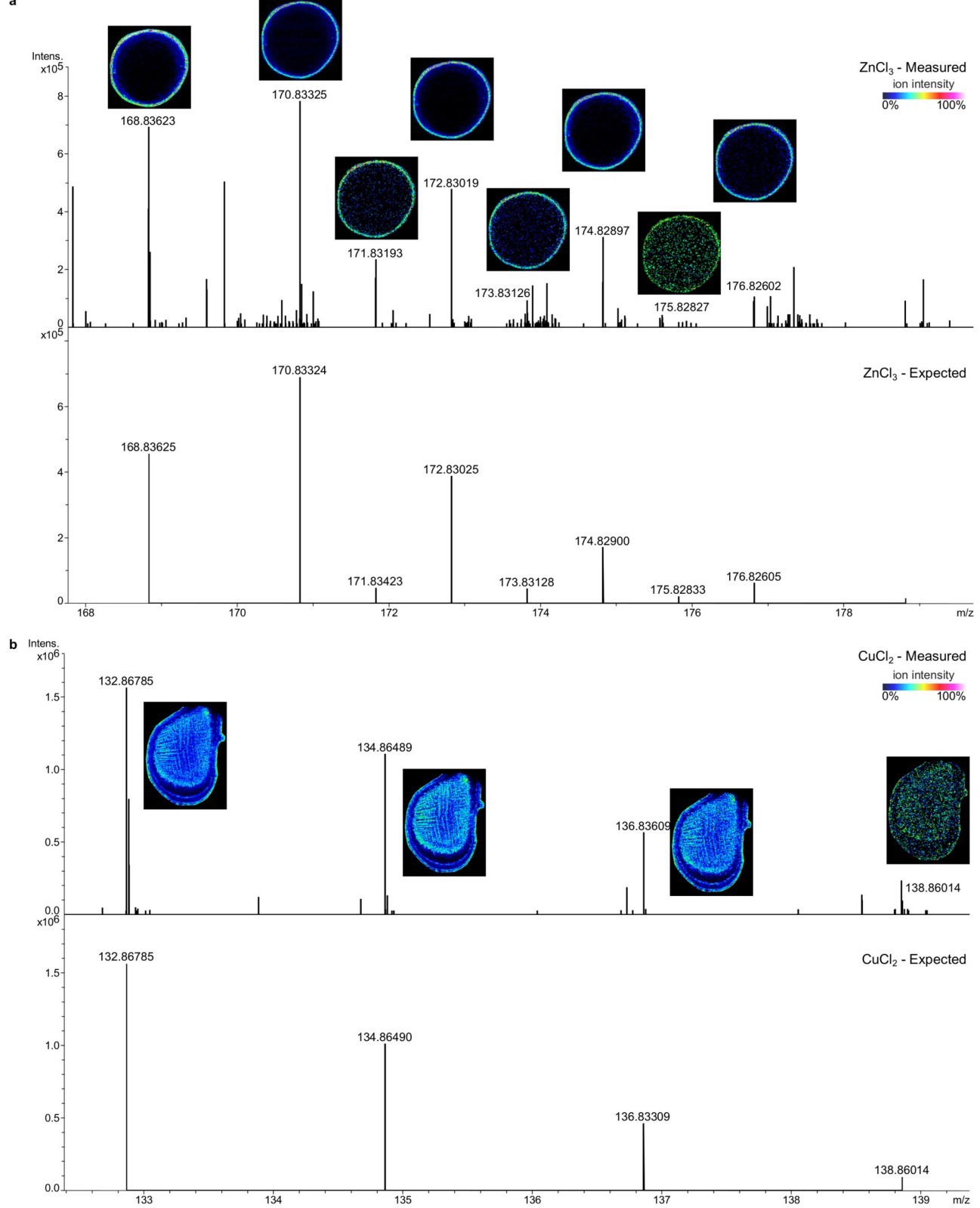

**Fig. 3 | Matrix assisted laser desorption ionization mass spectrometry imaging (MALDI-MSI) can be used to search for heavy metals in biosolids.** $ZnCl_3$ (**a**) and $CuCl_2$ (**b**) were successfully detected in biosolids. Measured spectra and expected spectra are displayed for $ZnCl_3$ in sample 2 and $CuCl_2$ in sample 6. Images are displayed with a 3 ppm window. $FeCl_3$, $PbCl_3$, $MgCl_3$ and $NiCl_2$ were also detected in the samples and are given in Supplementary Fig. 3. Three independent experiments were performed with similar results.

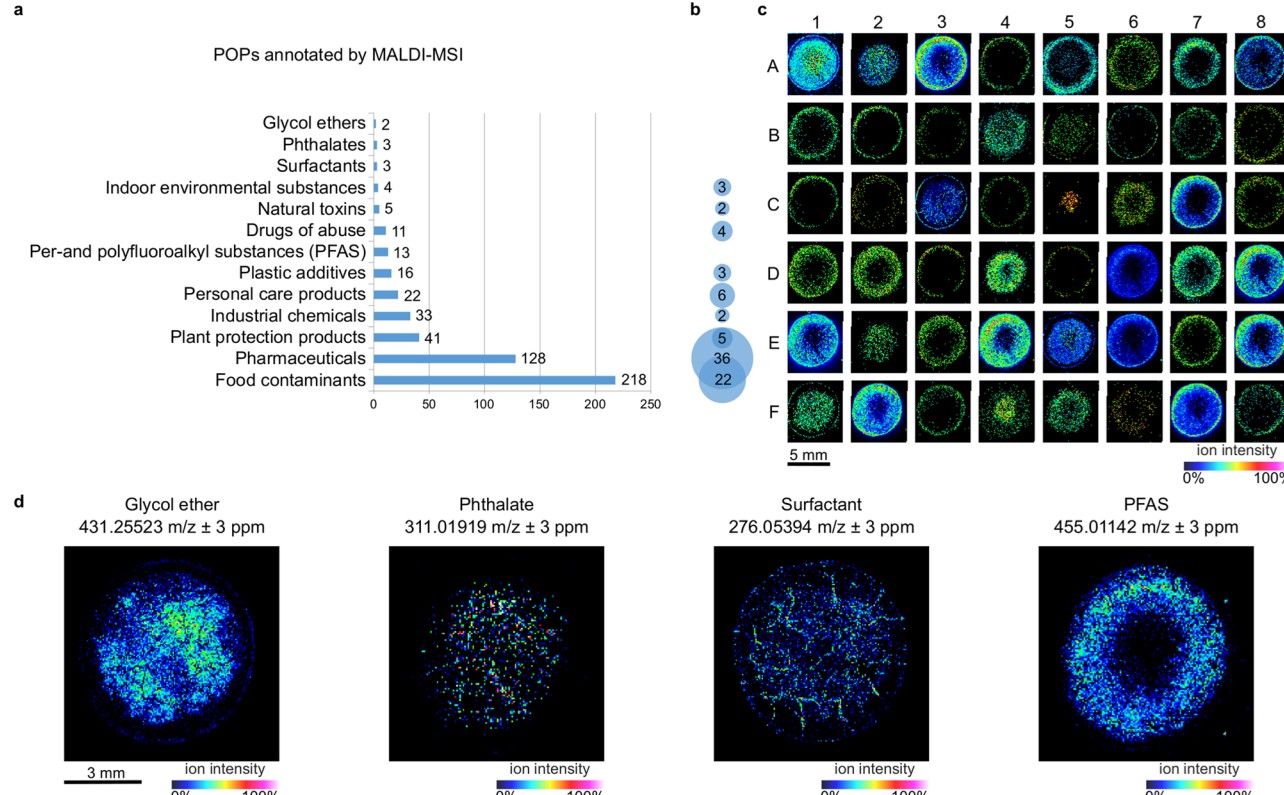

Fig. 4 | **MALDI-MSI allows for the annotation of hundreds of persistent organic pollutants (POPs) in biosolids. a** Number of POPs detected using matrix assisted laser desorption ionization mass spectrometry imaging (MALDI-MSI) sorted into families according to the Norman Database. Detailed annotations are available in Supplementary Data 1; **b** Number of POPs with a double annotation by MALDI-MSI and liquid chromatography coupled to high resolution tandem mass spectrometry (LC-HRMS/MS) or gas chromatography coupled to triple quadrupole tandem mass spectrometry (GC-TQ-MS/MS); **c** Example ion images of different POPs detected in biosolids: varied spatial distributions were observed among a single sample (representative distributions in sample 3 given as an example; detailed annotations given in Supplementary Table 7); **d** Spatial distribution of example POPs that were not detected by conventional methods. Glycol ether and surfactant in sample 4; phthalate and per-and polyfluoroalkyl substances (PFAS) in sample 3. The annotations presented in this figure are at level 3 of the Schymanski classification. Images are displayed with a 3 ppm window. Three independent experiments were performed with similar results. Source data are provided as a Source Data file.

## Spatial distribution of POPs

It has been described that the MALDI dynamics are lower than those in a conventional LC-MS analysis due to the absence of chromatographic separation. This point can be discussed here as we observed a specific distribution of the compounds on the samples, which is not comparable to what can be observed on-tissue where compounds are localized in specific cellular and extra cellular spaces due to biological reactions. The sample preparation and deposition procedures used for biosolids analysis induce a solvent-forced migration of the compounds. The spatial distributions we observed in biosolids seem to be strictly due to physico-chemical interactions of the compounds with the sample during deposition, acting as a form of in-sample chromatography, therefore obtaining retention areas comparable to a retention time in LC and allowing for a higher dynamic in MALDI-MSI. This migration is also comparable to what happens in thin layer chromatography (TLC), which was already successfully coupled to mass spectrometry imaging for the analysis of compounds of interest[56]. The structure of the samples, the presence of suspended solids, the water solubility, and the adsorption coefficient act on the dispersal of contaminants before the sample dries. As the solvent evaporates quicker than water, very polar compounds with low adsorption coefficients will tend to be pushed to the border of the samples with the liquid flow while apolar compounds with higher adsorption coefficients will be adsorbed to suspended solids that settle in the middle of the sample. The balance between water solubility and soil adsorption explains the various distributions which are intentionally amplified by creating a solvent-forced migration of the compounds during sample deposition. A slight accumulation of biosolids was measured on the outer edge of the deposits, explaining the presence of low soluble compounds with moderate soil adsorption in the periphery of the samples. The solvent-forced migration and the accumulation of up to 160 μL of sample on a single spot raises the sensitivity for low abundance compounds, which get concentrated in specific areas of the samples, and would not have been detected otherwise (Supplementary Fig. 4). A suitable ratio was found between the amount of sample deposited and the thickness of the spots to optimize sensitivity. Care must be taken not to deposit too much sample, which would decrease the conductivity needed for proper targeted and nontargeted MALDI-MSI analysis.

## Advantages and limitations of MALDI-MSI technique for the analysis of biosolids

MALDI-MSI is a complementary tool to the classical LC-MS/MS, GC-MS/MS, and ICP for the nontargeted analysis of POPs ad HMs in biosolids. The proposed method for sample preparation avoids the use of high amounts of dangerous solvents and high temperatures. It only requires a small amount of sample (1 g) and limited time in sample preparation. MALDI-MSI on biosolids can detect molecules that are not easily detected by classical methods, like the PFASs, which are extremely toxic POPs not otherwise easily analyzed. The method is aimed at a broad and interdisciplinary audience (soil science, engineering fields, agriculture and food security…), and might facilitate a cross-disciplinary dialogue with policymakers. Although very simple to apply, this method requires some attention to specific points: (i) biosolids have to be carefully homogenized before sampling 1 g, to avoid

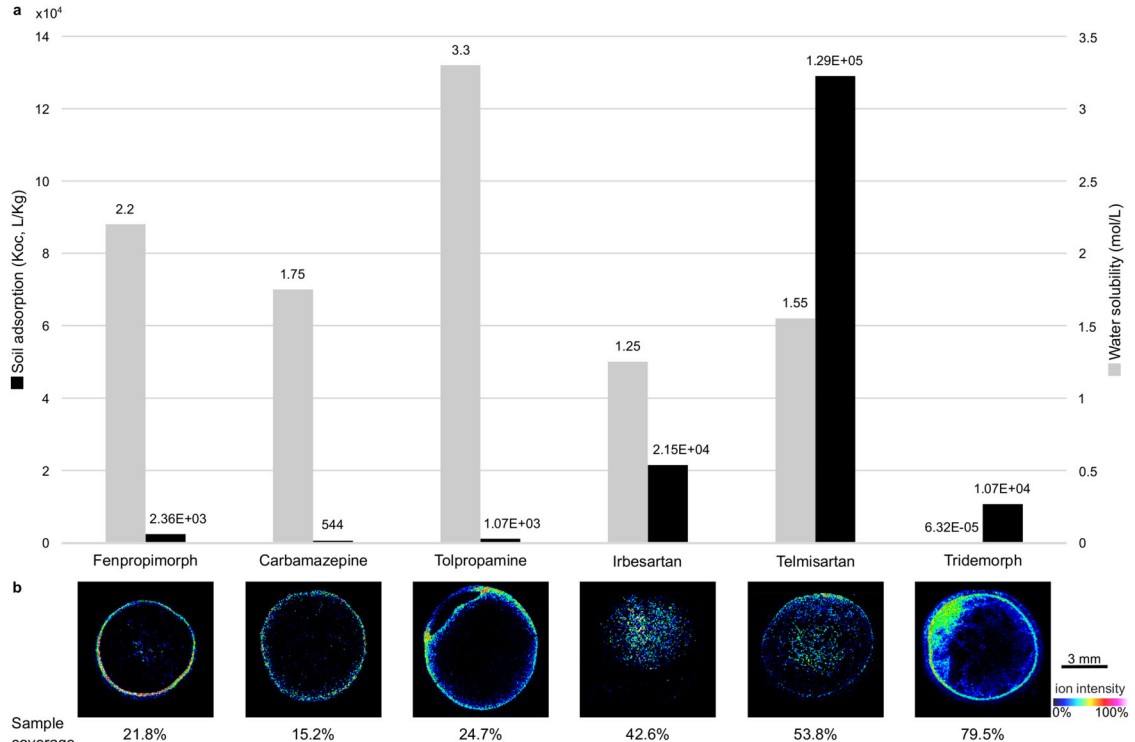

**Fig. 5 | Links between soil adsorption, water solubility and spatial distribution of persistent organic pollutants (POPs) annotated in biosolids (level 1 of the Schymanski classification). a** Soil adsorption and water solubility values were recovered from the CompTox Chemicals Dashboard; **b** Spatial distributions of the POPs in biosolid samples and sample coverage (% of the sample surface covered by the compounds of interest). Examples are given for fenpropimorph in sample 7, carbamazepine in sample 4, tolpropamine in sample 3, irbesartan in sample 4, telmisartan in sample 1, and tridemorph in sample 2. Images are displayed with a 3 ppm window. Three independent experiments were performed with similar results. Source data are provided as a Source Data file.

non-representative results; (ii) the quality of the copper tape used needs to be fine-tuned, using the most conductive tape as possible; (iii) sample deposition needs to be carefully done to keep a reasonable thickness and preserve conductivity; (iv) the data storage necessitates more informatics space than classical methods; (v) further developments need to be done in the annotation process, by developing MS/MS libraries of pathogen signatures and POPs and HMs in environmental contexts; (vi) as a new paradigm, the spatial distribution of POPs needs to be further deployed with a higher number of POPs in a higher number of biosolids and in different laboratories like MALDI Biotyper® is nowadays spread worldwide as a routine tool in hospitals for daily microbe screening and research.

### MALDI-MSI is a powerful tool for environmental sciences

MALDI-MSI is a dynamic field, and recent developments have increased sensitivity (MALDI$^2$), confirming the suitability of this technology for environmental science applications to detect low amounts of contaminants. The implementation of MALDI on rapid instruments reduces the acquisition time, allowing for very fast analysis. Data analysis and contaminants annotation is facilitated by recent advancements in commercially available softwares (SCiLS Lab coupled to MetaboScape), even in a nontargeted approach. Advances in the field of trapped ion mobility spectrometry (TIMS) will be very useful to differentiate isomers of contaminants that might have different biological activities/toxicities. Several studies have proven that MALDI-MSI is suitable for the detection of environmental POPs in biological samples[24,25]. Here, we applied MALDI-MSI to biosolids to detect and simultaneously spatially localize HMs and POPs. We proved that MALDI-MSI is a valuable tool for the analysis of biosolids before reuse, allowing for a safe, quick, and easy sample process to identify and quantify compounds of interest to achieve regulatory requirements.

## Methods

### Experimental design and sampling strategy

Biosolid samples were collected from nine wastewater treatment systems located in the Grand-Est region in France, representative of a large diversity of biosolids that can be used for soil improvement in agriculture. Three constructed wetland systems were sampled in Falkwiller (47.671816, 7.141041), Lutter (47.475279, 7.382069) and Montiers-sur-Saulx (48.541094, 5.276388). The management of constructed wetlands includes a cleaning of the filters on a regular basis, by collecting the organic matter which was deposited along with the incoming wastewater over time. Biosolids samples were collected in the first stage of the constructed wetlands as part of a land application scheme, in which the upper part of the filter containing organic matter is collected, placed on a tarp and naturally dried before uptake and land application. Other types of biosolids can be obtained from constructed wetlands[57] but were not covered in this study. These systems have capacities of approximately 1450, 970, and 160 people equivalents and were set up in 2009, 2010 and 2015, respectively. The average inlet flow rates were 450; 1080 and 190 m$^3$/day for Falkwiller, Lutter, and Montiers-sur-Saulx, respectively. Six activated sludge systems were sampled in Sarrebourg (48.75259104205218, 7.067368453111636), Hartzwiller (48.67461100734552, 7.0768876776777025) and Laneuville-lès-Lorquin (48.65444044102169, 7.01196463231784), La Wantzenau (48.631290, 7.832385), Meistratzheim (48.462222, 7.554734) and Sausheim (47.774990, 7.401391). These systems have capacities of approximately 37,000; 5,040; 9,133; 1,000,000; 504,550, and 490,000 people equivalent and were set up in 2001, 2005, 2011, 2006, 2011 and 2005, respectively. They are classical activated sludge systems, and the sludge from Sarrebourg and Laneuville-lès-Lorquin was treated by a filter press with the addition of lime (CaO). The sludge from Hartzwiller was filter pressed without lime to obtain biosolids. The sludge from La Wantzenau, Meistratzheim and Sausheim was used to produce biogas at 37 °C.

For Meistratzheim, biosolids were then dried at 180 °C. In La Wantzenau, only part of the sludge produced is used for biogas production. Then, sludge from thickening + biogas and sludge from thickening only are mixed together for the dehydration step. The sludge from Sausheim was subjected to a physico-chemical treatment (flocculation) and dried by rotatry drying. The average inlet flow rates were 8616; 1010; 1720; 350,322; 22,194; 126,115 m³/day for Sarrebourg, Hartzwiller and Laneuville-lès-Lorquin, La Wantzenau, Meistratzheim and Sausheim respectively. Representative homogenized biosolids accumulated over time were sampled. Five hundred grams of each biosolid used for soil improvement were collected in a glass bottle in February 2019 (samples 1–6) and May 2023 (samples 7–9) and transported at 4 °C before subsampling for HMs and POPs analysis.

## Chemicals

Acetic acid, formic acid and trifluoroacetic acid (TFA) were purchased from Sigma Aldrich (Missouri, USA). The extraction solvents (acetonitrile, methanol, isopropanol) were obtained from Fisher Chemicals (New Hampshire, USA) and were of analytical grade. Analytical-grade ethanol and ethyl acetate were obtained from Merck. Ammonium formate was acquired from Fluka Analytical (Missouri, USA), and NaOH was acquired from Agilent Technologies (California, USA). Deionized water was obtained from a Direct-Q UV (Millipore, Alsace, France) station.

The analytical standards telmisartan and melibiose were purchased from Sigma Aldrich; mannotetraose was obtained from Megazyme. Analytical standards of zinc and nickel were obtained from Merck; copper, iron, lead and magnesium were obtained from Sigma-Aldrich.

The MALDI matrices α-cyano-4-hydroxycinnamic acid (CHCA) and N-(1-naphthyl) ethylenediamine dihydrochloride (NEDC) were obtained from Bruker Daltonics (Bremen Germany) and Sigma Aldrich, respectively.

## MALDI-MSI analysis of biosolid samples

For the analysis of biosolid samples using MALDI-MSI, 1 g of homogenized biosolids was suspended in 2 mL of methanol/water (90/10). The samples were vortexed, and biosolid suspensions were deposited on an ITO-coated slide covered with conductive tape (conductive Cu tape, double-sided, N°796, Teraoka Seisakusho, Co Ltd.) previously described by Nakabayashi et al. for the analysis of delicate plant samples[39]. Other conductive tapes were used for preliminary tests (conductive Cu tape, double sided, N° 9711S-30; conductive C tape, double sided N° 77817-12 Electron Microscopy Sciences; Supplementary Table 2). Eight successive depositions of 20 μL on the same spot were performed for each sample, followed by drying steps under vacuum, to achieve a total of 160 μL deposited on a single spot for each sample replicate. Four biological replicates were deposited on the conductive tape, with one replicate dedicated to quantification. Samples were covered with MALDI matrix using an HTX M5 sprayer (HTX Technologies). For POPs and pathogen metabolite analysis, an α-cyano-4-hydroxycinnamic acid (CHCA) matrix was used at 10 mg/mL in 70% acetonitrile and 0.2% TFA. Samples were covered with 4 layers of matrix at a flow rate of 0.12 mL/min with a velocity of 1200 m/min and a spray temperature of 75 °C. For HMs analysis, an N-(1-naphthyl) ethylenediamine dihydrochloride (NEDC) matrix was used at 7 mg/mL in 70 °C ethanol as described by Andersen et al.[47]. Samples were covered with 14 layers of matrix at a flow rate of 0.06 mL/min with a velocity of 1200 mm/min and a spray temperature of 30 °C. Detailed matrix deposition parameters are given in Supplementary Table 5.

MALDI-MSI analysis was performed on a SolariX XR 7 T Fourier transform ion cyclotron resonance (FT-ICR) mass spectrometer over a mass range of 100–1000 Da. The analysis of POPs and pathogen metabolites was performed in positive ion mode with a magnitude size of 2 M. The instrument was calibrated before acquisition using the peaks of the CHCA matrix at m/z 379.0924, 399.0377, 401.04834, 417.04834, and 568.13505 in linear mode with a standard deviation below 0.2 ppm. Matrix peaks were used for online calibration during the acquisitions. Therefore, each spectra acquired was calibrated at the acquisitions as previously described by refs. 26,27. The MALDI parameters were set as follows: plate offset, 100 V; deflector plate, 200 V; laser power, 25%; laser shots, 200; frequency, 1000 Hz; laser focus, small; raster width, 50 μm; and average scan, 1. The analytical conditions were as follows: ion transfer capillary exit, 150 V; deflector plate, 200 V; funnel 1, 120 V; skimmer 1, 15 V; funnel RF amplitude, 150 Vpp; octopole frequency, 5 MHz; RF amplitude, 530 Vpp; quadrupole Q1 mass, 150 m/Z; collision cell collision voltage, −1 V; DC extract bias, 0.7 V; transfer optics time of flight, 0.600 ms; frequency, 6 MHz; RF amplitude, 350 Vpp; gas flow, 35%; para cell transfer exit lens, −20 V; analyzer entrance, −10 V; side kick, 0 V; side kick offset, −1.5 V; front trap plate 1.5 V; back trap plate, 1.5 V; back trap plate quench, −30 V; sweep excitation power, 22%; and ICR cell fills, 1.

The analysis of HMs was performed in negative ion mode with a magnitude size of 2 M over a 100–400 Da mass range. Due to the reduced number of NEDC matrix peaks, the instrument was calibrated before the acquisitions using the chlorinated peaks of HMs deposited on a sacrificial sample on the conductive tape at m/z 127.8735 ($NiCl_2$), 128.8921 ($MgCl_2$), 132.8678 ($CuCl_2$), 159.8451 ($MnCl_2$), 162.8424 ($NiCl_3$), 168.8362 ($ZnCl_3$), 183.8416 ($CdCl_2$) and 218.8104 ($CdCl_3$) in linear mode with a standard deviation below 0.2 ppm. The MALDI parameters were set as follows: plate offset, 100 V; deflector plate, 200 V; laser power, 30%; laser shots, 200; frequency, 1000 Hz; laser focus, small; raster width, 50 μm; and average scan, 1. The analytical conditions were as follows: ion transfer capillary exit, 150 V; deflector plate, 200 V; funnel 1, 120 V; skimmer 1, 15 V; funnel RF amplitude, 150 Vpp; octopole frequency, 5 MHz; RF amplitude, 530 Vpp; quadrupole Q1 mass, 150 m/Z; collision cell collision voltage, −1 V; DC extract bias, 0.7 V; transfer optics time of flight, 0.600 ms; frequency, 6 MHz; RF amplitude, 350 Vpp; gas flow, 35%; para cell transfer exit lens, −20 V; analyzer entrance, −10 V; side kick, 0 V; side kick offset, −1.5 V; front trap plate 1.5 V; back trap plate, 1.5 V; back trap plate quench, −30 V; sweep excitation power, 22%; and ICR cell fills, 1.

Fragmentation experiments can also be conducted in MALDI-MSI to go further in the annotation of POPs. An example is given with irbesartan (Supplementary Fig. 8), which identification was also confirmed using LC-HRMS/MS to level 1 of the Schymanski classification. Fragmentation was conducted after quadrupole isolation of m/z 429.24 ([M + H]+ adduct of irbesartan) in collision induced dissociation (CID) mode. The isolation window was set at 3 m/z, collision energy at 15 V, collision RF amplitude at 1600 Vpp and RF frequency at 2 MHz. MALDI parameters were adjusted to 45% laser power, 2000 laser shots, medium laser focus and 32 scans were manually acquired over the sample areas with the highest signal observed in MS.

## MALDI-MSI data processing

Raw data from MALDI-MSI acquisitions were imported from .mis files into SCiLS Lab MVS 2022a Premium 3D using the Import Wizard with instrument type « Bruker Magnetic Resonance Mass Spectrometry (MRMS) instruments », sqlite centroid spectra, automatic m/z range and automatic axis parameters. Segmentation was performed on the imported features with root mean square normalization and weak denoising to define the sample and off-sample regions. The sample region was exported in the .srd format suitable for import into MetaboScape 2021b for annotation. In MetaboScape, the peak list from the original .d data was loaded along with the .srd file from SCiLS Lab 2022a. Speckle size and n/Roll were chosen to obtain the highest coverage possible on the samples (Supplementary Table 6), and the intensity threshold was set at 500. The [M + H]+ was set as the primary ion; [M+Na]+, [M + K]+, [M + NH4]+ were set as seed ions. For POPs and pathogen metabolites, the obtained dataset was annotated using

suspect lists derived from the Norman Database (https://www.norman-network.com/nds/), Yeast Metabolome Database (YMDB, http://www.ymdb.ca/), *E. coli* Metabolome Database (ECMDB, https://ecmdb.ca/) and an internal database for targeted LC-HRMS analysis containing 2072 contaminants (848 pesticides and 1224 toxicants; Bruker Pesticides&Tox Screener). For HMs analysis, a suspect list of the expected chlorinated HMs ions was created and used for annotation. The annotations were performed with a threshold of maximum 3 ppm deviation on the exact mass of analytes, and a maximum of 30 for the mSigma value assessing the good fitting of isotopic profiles. The annotations obtained were imported back into SCiLS Lab 2022a with a 3 ppm width for visualization. The m/z images were manually curated to keep only the ions localized in the sample region as defined by the segmentation.

## Quantification of POPs and HMs

The quantification of copper and telmisartan in biosolid samples was performed using analytical standards and following previously published methods[47,58]. A standard curve was established by spotting 0.2 μL (telmisartan) and 1 μL (copper) of standard solutions on the ITO-coated glass slide to reach 2 to 15 pmol (telmisartan) and 0.5 to 10 μg (copper) on each spot. For each biosolid sample, one replicate was dedicated to quantification, and a spot of 15 pmol (telmisartan) and 10 μg (copper) was applied on the biosolid samples. During data processing, the signal obtained for each spot on the slide was recovered in SCiLS Lab 2022a and a standard curve was established using the quantification tool provided by the software. The on-sample signal for the 15 pmol/10 μg spots was measured, background signal from each sample itself was removed from the intensity of the signal measured on the 15 pmol/10 gμg spot, and extinction coefficients were calculated by comparing the intensity of the signal between the 15 pmol/10 μg spots on the slide and on each sample. From the extinction coefficients, standard curves were derived for each sample. Then, the intensity of the signal observed on the samples was reported on the recalculated standard curves to obtain quantification.

## Measurement of sample thickness

Sample thickness was measured using reflection-mode confocal laser scanning microscopy (LSM780, Carl Zeiss GmbH, Germany). Illumination was performed using a 10x/0.30 NA Plan-Neofluar objective lens, the 488 nm (0.3% AOTF) laser line and a T80/R20 beam splitter, after which light reflected on the exposed sample surface was imaged at zoom level 0.6. XZ optical sections were collected at 1 μm intervals.

## Measurement of physico-chemical parameters

In addition to the chemical and metabolic profiles, the physicochemical parameters of the biosolids were determined: pH, % organic matter, dry matter, organic carbon concentration, total Kjeldahl nitrogen (NTK), and ammonium (NH4). These parameters were analyzed by the standards NF EN 12176, NF EN 12879, NF ISO 14235, and NF EN 25663.

## Measurement of the metallic content of biosolid by ICP

The analysis of the HMs content in biosolids using conventional techniques was performed by Eurofins (Eurofins Analyses pour l'Environnement, Site de Saverne) following NF ISO guidelines. First, aqua regia mineralization was performed, then inductively coupled plasma (ICP) analysis was used for the quantification of Al, As, Cd, Ca, Cr, Cu, Mg, Ni, P, Pb, K, and Zn (NF EN 13346 method B, NF EN ISO 11885), and cold vapor atomic adsorption spectrometry (CV-AAS) was used for Hg (NF EN 13346 Method B, NF ISO 16722).

## LC-HRMS/MS analysis of biosolid samples

The method used for the extraction of contaminants and metabolites from the biosolid samples was adapted from ref. 26. Briefly, after biosolid sample homogenization, 10 g of each sample were weighed

and five replicates of each biosolid were processed. The samples were extracted for 16 h using 40 mL of methanol:$H_2O$ (90:10) and 1% acetic acid at 4 °C under continuous shaking with a magnetic stirrer. The samples were centrifuged for 15 min at 5242 g to collect the supernatant, which was freeze dried. A second extraction was performed under shaking at 4 °C with 20 mL of isopropanol:acetonitrile (90:10). The samples were centrifuged again and the supernatant was recovered and pooled with the first extraction to be freeze-dried. Samples were solubilized in 250 μL of MeOH:$H_2O$ (90:10) and diluted 10 times in $H_2O$ before analysis. Five biological replicates were prepared for each biosolid sample. The analysis was performed using liquid chromatography (LC) coupled to high-resolution mass spectrometry (HRMS) on a DioneX Ultimate 3000 (Thermo, Massachusetts, USA) coupled to a Q-TOF Impact II (Bruker, Germany). The samples were analyzed using both targeted and nontargeted screening approaches. For the targeted approach, the method allowed for the identification of POPs using a database containing 2072 contaminants (848 pesticides and 1224 toxicants; Bruker Pesticides&Tox Screener), their retention time, their exact mass and the exact mass of their fragments in broad-band collision-induced dissociation (bbCID) fragmentation mode. Samples were analyzed on a C18 column (AcclaimTM RSLC 120 C18, 2.2 μm, 120 A, 2.1 × 100 mm, Dionex bonded silica products) equipped with an Acquity UPLC- BEH C18 precolumn (1.7 μm, 2.1 × 5 mm) using a gradient of solvent A ($H_2O$:MeOH 90:10, 0.01% formic acid, 314 mg. $L^{-1}$ ammonium formate) and solvent B (MeOH, 0.01% formic acid, 314 mg. $L^{-1}$ ammonium formate) as described in detail by Villette et al. (2019) [28]. The spectrometer was operated in the positive ion mode using bbCID (for the targeted mode) and autoMS/MS (for the nontargeted mode) fragmentation modes on a mass range from 30 to 1000 Da, with a spectra rate of 2 Hz (bbCID) or 8 Hz (AutoMS/MS). Analytical quality checks were performed using a mix of pesticides to assess the retention times, exact mass and mass of the fragments (refs 31972 and 31978, Restek).

## LC-HRMS/MS data processing

POPs and metabolites detected in the samples using MALDI-MSI were searched for in the LC-HRMS/MS raw data to confirm their identifications. The compounds from which a retention time was confirmed were identified to level 1 of the Schymanski classification[40]. For the nontargeted method, the annotations were performed using Metaboscape 4.0 (Bruker), which creates a bucket containing all adduct forms for a mass of interest, which corresponds to level 5 (exact mass) in the Schymanski classification. To obtain the dataset, an intensity threshold of 1000 was set up. The annotations were achieved using the criteria of mass deviation (Δm/z) under 3 ppm and a mSigma value (assessing the good fitting of isotope patterns) under 30. Tentative identifications were achieved using suspect lists derived from the annotated compounds found in MALDI-MSI datasets. For the targeted analysis, the annotations were performed using TASQ 1.4 (Bruker), resulting in identifications to level 1 of the Schymanski classification. The selection criteria were a signal-to-noise ratio higher than 3, a retention time variation lower than 0.3 min, a mSigma value under 30, an exact mass variation lower than 3 ppm and matching fragment ions.

## GC-TQ-MS/MS analysis of biosolid samples

Samples were prepared as described for LC-HRMS/MS analysis, a 50 μL aliquot was freeze-dried and solubilized in 50 μL ethyl acetate for GC-TQ-MS/MS analysis. Gas chromatography (GC) was performed on a triple quadrupole detector (TQ, SCION 436-MS, Bruker) in multiple reaction monitoring (MRM) mode to obtain fragmentation of the compounds (MS/MS) as already described by ref. 59. A Rxi®−5Sil MS column (Restek, 30 m, 0.25 mm ID, 0.25 μm) was used, with a constant flow of helium gas (1 mL/min, Alphagaz 2, Air Liquide). The injector was operated at 280 °C with a pressure pulse at 30 psi, the oven was set at 70 °C for 0.7 min and the temperature was increased to 180 °C at

30 °C/min and then to 300 °C at 10 °C/min, with a stabilization time of 2 min. The electron impact energy for fragmentation was 70 eV, and 328 POPs (pesticides, toxicants, and polycyclic aromatic hydrocarbons) were targeted. At least two daughter ions were recorded for each analyte, and a retention window of 5 min was used. Analytical quality checks were performed using a mix of pesticides to assess the retention times (ref 32563, Restek).

## GC-TQ-MS/MS data processing

Data processing was performed in MS Data Review 8.2 software (Bruker). POPs were identified to level 1 of the Schymanski classification[40] based on their retention time, mass of the parent ions and presence of at least 2 daughter ions. The signal-to-noise ratio (S/N) was set to 3, the retention time window was set to 0.25 min, and the mass variation was set to 0.5 Da.

## Reporting summary

Further information on research design is available in the Nature Portfolio Reporting Summary linked to this article.

## Data availability

The mass spectrometry data and physico-chemical data generated in this study are provided in Supplementary Information and Supplementary Data 1. Source Data are provided with this paper. The MALDI imaging data generated in this study have been deposited in the MetaSpace database under https://metaspace2020.eu/project/villette-2023-nat-commun. Source data are provided with this paper.

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

## Acknowledgements

The authors would like to acknowledge the Agence de l'Eau Rhin Meuse, the villages of Lutter and Falkwiller, the Communauté de communes de Moselle Sud and Portes de Meuse, the facilities of La Wantzenau, Meistratzheim and Sausheim for access to biosolids from the constructed wetland and activated sludge systems. The authors also wish to thank the Laboratoire d'Etude des Eaux (LEE) for its help. L.M. discloses support for the research of this work from Agence de l'Eau Rhin Meuse (AERM) [grant number 183696]. Figures 1a and 2a were created with BioRender.com with full license to publish.

## Author contributions

C.V. performed the LC-HRMS/MS and MALDI-MSI analyses, processed the MALDI-MSI data, prepared the figures and wrote the manuscript; L.M. sampled the biosolids, performed the sample preparation for LC-HRMS/MS analysis, processed the LC-HRMS/MS data and discussed the manuscript; J.Z. performed the GC-TQ-MS/MS experiments; J.M. performed the confocal laser scanning microscopy experiment; A.W. discussed the manuscript; D.H. designed and supervised the study, performed the MALDI-MSI sample preparation, discussed the results and wrote the manuscript.

## Competing interests

The authors declare no competing interests.
