## [Peer Review File · Nature Communications]

Mass spectrometry imaging for biosolids characterization to assess ecological or health risks before reuseEditorial Note: This manuscript has been previously reviewed at another journal that is not operating a transparent peer review scheme. This document only contains reviewer comments and rebuttal letters for versions considered at *Nature Communications*.

REVIEWER COMMENTS

Reviewer #1 (Remarks to the Author):

The authors have successfully addressed the reviewers concerns that were raised including the addition of new figures and data that highlights the importance of MSI rather than just MS in the data collection strategy. I now recommend this article for publication.

Reviewer #3 (Remarks to the Author):

The manuscript describes a MALDI-MSI method for detection of contaminants in biosolids. While the results are interesting and the method has potential, I do not see the novelty or impact of this paper reaching the bar for publication in *Nature Communications*. My rationale for this recommendation is that while detection of contaminants in biosolids is of importance, this method represents an interesting, but ultimately incremental advance over existing LC and GS-MS/MS methods for non-target analysis. Additionally, the spatial distribution information acquired through MALDI-MSI is of limited useful in the biosolids application that is the focus of this manuscript.

I also have a question about the treatment of the constructed wetlands biosolids samples. The authors correctly describe biosolids as distinct from sewage sludges and having advanced treatment to support land application, however, the authors only describe the treatment in Table 1 as 'drying on a tarp'. Was this done by the authors or are these materials dried in this manner as part of a land application scheme? It is not clear if these samples are truly biosolids or rather a sediment or sludge obtained from the wetland and the paper should reflect this.

Additionally, the biosolids samples do not necessarily represent diversity across wastewater treatment technologies or sludge treatment and stabilization. The treatment plants were either constructed wetlands with very small population equivalents, or relatively small activated sludge treatment systems. All of the biosolids samples appear to be generated from a system without any thermal treatment. Again, while the method proposed is interesting, the authors did not examine an exhaustive or representative set of biosolids samples

I do agree with the author that the method and the spatial data that could be obtained could be interesting for environmental samples, however, beyond some incremental advances in sample preparation, I do not see a significant advancement represented here

REVIEWER COMMENTS

Reviewer #1 (Remarks to the Author):

The authors have successfully addressed the reviewers concerns that were raised including the addition of new figures and data that highlights the importance of MSI rather than just MS in the data collection strategy. I now recommend this article for publication.

We thank you for your time and effort in reviewing the proposed manuscript.

Reviewer #3 (Remarks to the Author):

The manuscript describes a MALDI-MSI method for detection of contaminants in biosolids. While the results are interesting and the method has potential, I do not see the novelty or impact of this paper reaching the bar for publication in Nature Communications.

The impact of our study stands in the fact that the social acceptance of biosolids reuse requires a methodology to control their quality. The method we propose is aimed at a broad and interdisciplinary audience (soil science, engineering fields, agriculture, and food security...), and might facilitate a cross-disciplinary dialogue with policymakers. Indeed, the environmental risk assessment is necessary before reuse of biosolids in land application or in other uses. A sustainable sludge management procedure must include a rapid method, which can be used on a daily basis to control the quality of the biosolids produced. This is in line with the continuous efforts made to ameliorate wastewater management facilities.

To clearly explain the novelty, until now a single method for the analysis of biosolids did not exist, at least three different methods were necessary for the analysis of POPs and HMs, namely LC-MS/MS, GC-MS/MS and ICP. This multiplicity of methods necessitates to multiply the analysis time and sample amounts and this is a major drawback which prevents the analysis of a large series of samples. Here, we propose a unique method for the analysis of POPs and HMs, which significantly reduces the time for sample preparation and the needed amount of biosolids, only 1g. Moreover, the proposed method is nontargeted, which means that there is no limitation in terms of POPs families, and the obtained data can be re-interrogated later on if new POPs or HMs are to be searched.

Our study shows for the first time a large screen of POPs and HMs in biosolids using MALDI ionization source. This has never been done before. In the literature, you can only find studies made on one or a few biosolids with mostly targeted POPs only. The novelty relies also on the use of a specific MALDI matrix that chemically modifies HMs, which ends up by an increase of HMs ionization and enables their analysis by mass spectrometry. This technic recently published for HMs analysis *in situ* in biological samples was here adapted for the first time in non-biological material.

HMs are not easily analyzed by LC-MS/MS and GC-MS/MS, therefore the ICP technique is used for HMs screening. By using MALDI-MSI with a specific ionization matrix, we

open a new possibility to analyze HMs in solid environmental samples in a very quick and simple way.

The novelty here in our work is that we introduce MALDI-MSI as a solution to overcome a major drawback that explains why MALDI is not used in complex sample analysis. Indeed, the major drawback of MALDI is not linked to its ionization capacities (quite the contrary) but to the absence of chromatography coupling possibility. No separation of the molecules is possible before ionization, which lowers the dynamic range of MALDI. In comparison, electrospray and electronic impact ionizations are both possibly, easily, coupled to chromatography and therefore well adapted for complex sample analysis. MALDI ionization is, like electrospray, a soft ionization technic, very sensitive and adapted to a large spectrum of molecules with the advantage to yield mostly monocharged ions (simple spectral interpretation). The novelty here in our study relies on the fact that we overcome this MALDI major drawback (absence of separation technic coupling). **Indeed, by using a solvent forced extraction method, we forced molecules (POPs and HMs) to migrate differentially along the biosolid and to concentrate, based on at least two physico-chemical parameters (water solubility and soil adsorption).** This molecular migration brings a slight separation step of the molecules comparable to a “chromatography” that is in any case enough to obtain a slight molecular separation that increases ionization efficiency, so more POPs and HMs identified.

The novelty of our work does not only rely on the technical approach but also on the results obtained. Indeed, our work represents the first large screen of POPs and HMs made on biosolids independently of the analytical technique used. In the literature, no equivalent POPs and HMs screening can be found, neither by LC-MS/MS nor by GC-MS/MS or ICP.

The novelty relies also on the fact that important POPs families, which are known to be difficult to detect with classical methods, are detected using the proposed method. Examples are shown with PFASs, surfactants, phthalates and glycol ethers in Fig. 4.

Additional experiments were suggested by Reviewer #3 to increase the quality of the proposed manuscript. All the suggestions from Reviewer #3 were followed, three samples were added, and complementary experiments were performed. With the revisions we performed following Reviewer #3 comments, we believe that our manuscript now reaches the bar for publication in Nature Communications.

My rationale for this recommendation is that while detection of contaminants in biosolids is of importance, this method represents an interesting, but ultimately incremental advance over existing LC and GS-MS/MS methods for non-target analysis.

As explained above, MALDI-MSI is a complementary tool to the classical LC-MS/MS, GC-MS/MS and ICP for the nontargeted analysis of POPs and HMs in biosolids. The proposed method for sample preparation avoids the use of high amounts of dangerous solvents and high temperatures. It only requires a small amount of sample (1g) and limited time in sample preparation. The method is aimed at a broad and interdisciplinary audience (soil science, engineering fields, agriculture and food security...), and might facilitate a cross-disciplinary dialogue with policymakers. **The advantages of the proposed method are clear, with a reduced sample preparation time, a reduced amount of sample needed and a faculty to detect POPs families that are difficult to detect with classical methods. The added value of the proposed method stands in the spatial distribution of POPs in the samples, which is correlated with the physico-chemical properties of the studied POPs.** This cannot be

discovered with any other method and brings information regarding water solubility and adsorption coefficient of the POPs. Moreover, in the absence of chromatography, the distribution of the pollutants over the sample surface also serves as a separation technique to widen the dynamic of the MALDI source. The proposed method gives a molecular image of biosolids which could be used as a reference database in case an issue in the wastewater treatment process is suspected. The molecular image can also serve as a reference regarding the quality of the biosolids used.

Additionally, the spatial distribution information acquired through MALDI-MSI is of limited useful in the biosolids application that is the focus of this manuscript.

In biological tissues, the spatial distribution given by MALDI-MSI highlights cellular and tissular molecular profiles due to biological reactions. MALDI-MSI also proved useful in other fields: in forensics to reconstitute fingerprints or search for drugs of abuse in low amounts of hair; in art to study old paintings without invasive protocols, in materials analysis to search for surface irregularities. As in the cited applications, we demonstrate the ability of MALDI-MSI to bring additional information on the samples of interest without a biological background.

In the case of biosolids analysis in the context of sustainable sludge management and environmental and health risk assessment, the spatial distribution acquired through MALDI-MSI is of high importance for the detection of POPs and HMs, which would not get concentrated enough otherwise.

The spatial distribution reflects the heterogeneous physico-chemical properties of the studied biosolids, which are all specific due to the processes used in wastewater treatment facilities to obtain these materials. The spatial distribution in biosolids highlights POPs and HMs distributions due to physico-chemical exchanges between the compounds, the biosolids and the solvents used for the extraction, which could not be seen and understood otherwise and as easily as with MALDI-MSI.

To better understand the utility of the spatial distribution information acquired through MALDI-MSI, what would have happened if we had used MALDI (no spatial distribution information) instead of MALDI-MSI?

- 1- We would not have shot the samples on the whole surface, which means that we would have missed the heterogeneous profiles of the samples (specific areas where POPs or HMs are concentrated)
- 2- We would have detected no or a small number of POPs due to the fact that some POPs would have been concentrated in specific areas, which might not have been shot
- 3- We might have considered that our method was not reproducible, as we would have detected some pollutants in specific areas of the samples, but not in other areas, without understanding why (concentration effect)
- 4- If shooting the whole sample surface by MALDI only, we would have been unable to understand the heterogeneity of the samples, without an image to “reconstruct” the specific areas where pollutants get concentrated by the solvent-forced extraction which happens during deposition.

I also have a question about the treatment of the constructed wetlands biosolids samples. The authors correctly describe biosolids as distinct from sewage sludges and having advanced treatment to support land application, however, the authors only describe the treatment in Table 1 as 'drying on a tarp'. Was this done by the authors or are these materials dried in this manner as part of a land application scheme? It is not clear if these samples are truly biosolids or rather a sediment or sludge obtained from the wetland and the paper should reflect this.

Reviewer #3 is right, there are several ways to obtain biosolids from constructed wetlands, as described by Uggetti *et al.*, 2012. **In our case, the material was not dried by the authors but as part of a land application scheme monitored by the persons in charge of the constructed wetlands facilities.** A better explanation of the way biosolids were obtained from constructed wetlands was added line 402-408: "The management of constructed wetlands includes a cleaning of the filters on a regular basis, by collecting the organic matter which was deposited along with the incoming wastewater over time. Biosolids samples were collected in the first stage of the constructed wetlands as part of a land application scheme, in which the upper part of the filter containing organic matter is collected, placed on a tarp and naturally dried before uptake and land application. Other types of biosolids can be obtained from constructed wetlands⁵⁷ but were not covered in this study."

Additionally, the biosolids samples do not necessarily represent diversity across wastewater treatment technologies or sludge treatment and stabilization. The treatment plants were either constructed wetlands with very small population equivalents, or relatively small activated sludge treatment systems. All of the biosolids samples appear to be generated from a system without any thermal treatment. Again, while the method proposed is interesting, the authors did not examine an exhaustive or representative set of biosolids samples.

Reviewer #3 is right; we extended our sampling to bigger sites (1,000,000; 490,000 and 204,550 people equivalent). The 1,000,000 site is the second biggest in France. It is to be noted that constructed wetlands are never used over 2,000 people equivalent, therefore the bigger sites were activated sludge types. Reviewer #3 is right, we added biosolids obtained from systems using thermal treatments: either mild thermal treatment (37°C) or hard thermal treatment (180°C).

Therefore, we performed further experiments and a substantial amount of further analysis and revisions following the comments made by the Editor and Reviewer #3. We are happy to let you know that the obtained results strengthen the applied potential of the proposed method.

Indeed, the three new samples that we included in our study now show that our method is also compatible for POPs and HMs screening in biosolids coming from very big wastewater treatment facilities (up to 1,000,000 people equivalent) and biosolids thermally treated.

If we sum up, now our study deals with samples that are representative of a large diversity of biosolids coming from the two dominant wastewater treatment facilities in developed countries that are activated sludge and constructed wetland. Samples in our study are also representative of the different processes used to obtain biosolids from sludges: drying, lime stabilization, digestion... These processes are impacting qualitatively and quantitatively the POPs and HMs contents in the obtained biosolids. Interestingly, in all cases, biosolids could be analyzed with

our methodology. Even biosolids resulting from biogas production and heating at 180°C, expected to contain lower amounts of POPs and HMs, were successfully analyzed using the proposed method, proving undoubtedly that any kind of biosolid can be analyzed using our method.

We added some nuance in our speech

-in the sampling strategy line 109-111: “The sampling strategy consisted of choosing very diversified biosolids that are representative of the most used WWTP (Table 1, Supplementary Table 1, Supplementary Fig. 1), **although the presented sampling is not exhaustive due to the variety of treatments existing.**”

-in the discussion, “all biosolids” was replaced by “varied types of biosolids line 281-283: “In doing so, we demonstrate that this method is suitable for the analysis of **varied types** of biosolids and could be extended to other powdered materials (soil, sand, compost, etc.).”

With these additions (bigger sites and thermal treatment), we believe that our manuscript now gives a better overview of the different types of biosolids (mostly used) although we do not claim to be exhaustive due to the multiplicity of the existing types of treatments.

I do agree with the author that the method and the spatial data that could be obtained could be interesting for environmental samples, however, beyond some incremental advances in sample preparation, I do not see a significant advancement represented here

We thank Reviewer #3 for the comment about the utility of the method and the spatial data obtained in the context of environmental samples analysis. We performed all the incremental advances recommended by Reviewer #3 by adding three biosolids samples to extend the covered diversity across wastewater treatment technologies or sludge treatment and stabilization. Indeed, we added three big activated sludge systems (1,000,000; 490,000; 204,550 people equivalent) among which two are using mild thermal treatment (37°C) and one is using harsh thermal treatment (180°), as suggested by Reviewer #3. Table 1, Fig. 1, Fig. 5, Supplementary Fig. 1, Supplementary Fig. 2, Supplementary Table 1, Supplementary Table 3, Supplementary Table 6 and Supplementary Dataset 1 and methods section were modified accordingly.

With these substantial amounts of additional information, the obtained results strengthen the applied potential of the proposed method. We believe that the proposed manuscript now reaches the expected quality for publication in Nature Communications.

REVIEWERS' COMMENTS

Reviewer #3 (Remarks to the Author):

After the re-review of the authors response and the updated manuscript, I am pleased to recommend publication.